# A Novel Loss-of-Function *SEMA3E* Mutation in a Patient with Severe Intellectual Disability and Cognitive Regression

**DOI:** 10.3390/ijms23105632

**Published:** 2022-05-18

**Authors:** Alyssa J. J. Paganoni, Federica Amoruso, Javier Porta Pelayo, Beatriz Calleja-Pérez, Valeria Vezzoli, Paolo Duminuco, Alessia Caramello, Roberto Oleari, Alberto Fernández-Jaén, Anna Cariboni

**Affiliations:** 1Department of Pharmacological and Biomolecular Sciences, Università degli Studi di Milano, 20133 Milan, Italy; alyssa.paganoni@unimi.it (A.J.J.P.); federica.amoruso@unimi.it (F.A.); roberto.oleari@unimi.it (R.O.); 2Genologica Medica, 29015 Malaga, Spain; jporta@genologica.com; 3Pediatric Primary Care, C. S. Doctor Cirajas, 28017 Madrid, Spain; beacalleja@telefonica.net; 4Department of Endocrine and Metabolic Diseases and Laboratory of Endocrine and Metabolic Research, IRCCS Istituto Auxologico Italiano, 20145 Milan, Italy; v.vezzoli@auxologico.it (V.V.); p.duminuco@auxologico.it (P.D.); 5UK Dementia Research Institute, Imperial College London, London SW7 2AZ, UK; a.caramello@imperial.ac.uk; 6Neuropediatric Department, Hospital Universitario Quirónsalud, School of Medicine, Universidad Europea de Madrid, 28670 Madrid, Spain; 7Department of Pediatric Neurology, Hospital Universitario Quirónsalud, 28223 Madrid, Spain

**Keywords:** semaphorin 3E, intellectual disability, cognitive regression, neurodevelopmental disorder

## Abstract

Intellectual disability (ID) is a neurological disorder arising from early neurodevelopmental defects. The underlying genetic and molecular mechanisms are complex, but are thought to involve, among others, alterations in genes implicated in axon guidance and/or neural circuit formation as demonstrated by studies on mouse models. Here, by combining exome sequencing with in silico analyses, we identified a patient affected by severe ID and cognitive regression, carrying a novel loss-of-function variant in the semaphorin 3E (*SEMA3E*) gene, which encodes for a key secreted cue that controls mouse brain development. By performing ad hoc in vitro and ex vivo experiments, we found that the identified variant impairs protein secretion and hampers the binding to both embryonic mouse neuronal cells and tissues. Further, we revealed SEMA3E expression during human brain development. Overall, our findings demonstrate the pathogenic impact of the identified *SEMA3E* variant and provide evidence that clinical neurological features of the patient might be due to a defective SEMA3E signaling in the brain.

## 1. Introduction

Intellectual disability (ID) is a lifelong disorder that most likely arises from early neurodevelopmental defects and is characterized by subaverage intellectual and adaptive functioning due to abnormalities of brain structure and function, whose onset occurs before the age of 18 [1]. The DSM-V (Diagnostic and Statistical Manual of Mental Disorders) characterizes the severity of ID in four levels, from mild to profound, based on a person’s adaptive functioning and on the amount of a support that a person needs. Multiple factors are involved in the etiology of neurodevelopmental defects characterized by ID and the heterogeneity makes genetic and clinical diagnosis challenging. Genetic factors include genetic variations such as aneuploidies, copy number variations (CNVs), and tandem repeats in specific genes [2]. However, the most frequent form of ID is the Down Syndrome. Among the environmental factors, prenatal exposure of the fetus to toxic substances (e.g., alcohol, UV radiation) and infections during pregnancy (e.g., rubella, cytomegalovirus) are reported to cause ID. In addition, multiple problems during or after birth, such as lack of nutrition, may cause brain damage, leading to ID [3]. Although the etiological factors of ID are very broad, half of the cases are still unexplained [4].

Among them, recent evidence suggests that mutations of genes encoding for key molecules controlling neuron migration, axon guidance, or neural circuit formation during brain development might be involved [5,6]. The appropriate development and maturation of neural circuit strongly relies on the tightly coordinated action of long- and short-range axon guidance cues. In this context, semaphorins (SEMAs) are fundamental players in mediating cell–cell communication and controlling a wide variety of cellular functions [7,8]. Although originally discovered as repelling signals for growing axons, these guidance molecules have been shown to play diverse developmental functions that shape the nervous system, including regulation of neuronal cell migration, dendritic arborization, apoptosis, synaptic plasticity, and targeting [7,9,10]. Transmembrane plexin (PLXN) proteins represent predominant high affinity SEMA receptors [11]. In addition, several co-receptors also associate with SEMA receptors and have profound effects on the signaling outcome upon SEMA binding. These proteins directly bind SEMAs and initiate signaling (e.g., integrins), act as ligand binding co-receptors (e.g., neuropilins (NRPs)), and/or work as part of multimeric receptor complexes (e.g., receptors tyrosine kinases (RTKs), such as vascular endothelial growth factor receptor (VEGFR)) [12,13].

Provided their many roles are in brain development and function, SEMAs and their related receptors have been implicated in both developmental and adult-onset nervous system diseases [13,14]. In particular, mutations in members of the semaphorin–plexin signaling have so far mainly been linked to GnRH deficiency (GD), which could be due to the defective development or function of hypothalamic GnRH neurons that control the reproductive axis [8]. Further, emerging evidence strongly supports that this class of molecules might also be implicated in the pathogenesis of broader neurodevelopmental disorders (NDDs). This has been highlighted by association of chromosome microdeletions, including *SEMA5A* and *SEMA7A* genes with autism spectrum disorder (ASD) and co-diagnosed ID [15,16], as well as by recent studies unveiling links between *PLXNA3*, *PLXNA2*, and *PLXNA1* receptor variants with NDD syndromes [6,17,18].

Within the semaphorin family, the secreted class 3 Semaphorin 3E, encoded by the *SEMA3E* gene (chr. 7q21.11), regulates key functions during mouse brain and vascular development [19,20]. Unlike other class 3 SEMAs, SEMA3E can exert a biological effect by directly binding, with high affinity, PLXND1 receptor. Nevertheless, different co-receptors, including NRP1 and VEGFR2, have been shown to modulate SEMA3E/PLXND1 signaling [21,22,23]. Studies in knockout (KO) mice revealed that in the brain, SEMA3E, via its PLXND1 receptor, regulates the migration of cortical hem-derived Cajal Retzius cells [24] and of newborn neurons in the post-natal olfactory bulb [25]. SEMA3E is also required for the patterning of several axonal tracts, such as subiculo-mamillary [21] and entorhino–hippocampal axons [26]. In addition, SEMA3E has been found to play a role in the survival of hypothalamic GnRH neurons that control the reproductive axis [27]. Accordingly, missense mutations in this gene have so far been found in a few patients affected either by Kallmann or by CHARGE syndrome, which share GD and consequent reproductive defects [14,27]. Yet, although *Sema3e*-null mice display several brain abnormalities that could be associated with ID in humans, no reports of *SEMA3E* mutations in human patients displaying similar phenotypes to have been reported.

In recent years, the advent of novel forms of genetic testing helped in the identification of genetic causes in patients with unexplained ID. However, the molecular and circuit mechanisms underlying the pathophysiology of this disorder remain elusive and, consequently, effective treatments have not yet been established. Thus, for effective early interventions, it becomes important to investigate the specific biological and molecular causes of ID in each patient.

Here, we report for the first time a 19-year-old male patient affected by cognitive regression and severe ID, and carrying a novel frameshift *SEMA3E* mutation, whose functional relevance was dissected by applying tailored in vitro and ex vivo models. Our results reveal the loss-of-function nature of the identified mutation, which prevents SEMA3E from exerting its biological functions during early brain development and plausibly explains the patient’s neurological signs.

## 2. Results

### 2.1. Clinical Features of the Patient

A 5-year-old male, a second child from non-consanguineous healthy parents of Spanish origin, was referred to our clinic. Partum occurred via uncomplicated caesarean section after 39-weeks of pregnancy. Apgar scores were 9–10 at 1 and 5 min, respectively. Birth weight was 2950 g. Family history was not relevant; his parents and his sister were healthy.

He started walking unsupported at 13 months, but his initial language and communication abilities were delayed; the first bisyllabic words occurred at 30 months. At the age of 5 years, he used some words and short sentences.

The clinical examination disclosed a weight of 19 kg (35th centile), a height of 110 cm (30th centile), and an OFD of 51.2 cm (50th centile), without dysmorphic features; neurological examination was normal, including cranial nerve exam, tone, DTR, and sensory and coordination abilities. Observation of the patient revealed verbal and nonverbal communication deficits.

Neuropsychological evaluation revealed an intellectual and developmental quotient of 83 and 70 according to the Weschler Preschool and Primary Scale of Intelligence III and Cumanin respectively. Specific problems in executive functions, attention and language abilities were demonstrated in this study (<1st centile according to Peabody Picture Vocabulary Test and Illinois Test of Psycholinguistic Abilities)

Routine laboratory screening including thyroid function and neurometabolic tests were within the normal range. Brain MRI, sleep video-EEG test and auditory evoked potentials displayed normal results.

Conventional genetic studies (karyotype and array-based comparative genomic hybridization with a 400 k custom array) revealed no abnormalities (Table 1).

At the age of 10 years, complex motor tics appeared, associated to atypical behaviors (checking doors, repetitive table tapping, etc.). During adolescence, conceptual problems worsened, presenting problems in social adaptation, excessive shyness, and less autonomy for daily activities. This worsening was not related to psychological factors or medical treatments.

At the ages of 7, 10, and 19 years, different cognitive evaluations using Weschler Scales (WISC-IV and WAIS, according to his age) revealed an IQ of 63, 50, and 38, respectively (Table 2). At 10 years, severe executive and attention problems were registered (scores <1st centile in Continuous Performance Tests, Test of Memory and Learning (Tomal-2), Rey complex figure test, and Neuropsychological Evaluation of Executive Functions in Children, ENFEN (Evaluación Neuropsicológica de las Funciones Ejecutivas en Niños). Adaptative skills were evaluated with the Adaptive Behavior Assessment System II at the age of 19, showing severe problems in all domains (<5th centile in conceptual, social, practical, and general domains).

According to this evolution, the observed cognitive regression and the presence of a severe intellectual disability, the medical study was completed with WES in trio.

### 2.2. A De Novo Frameshift SEMA3E Mutation in the Patient

Whole exome sequencing (WES) trio analysis revealed a de novo *SEMA3E* variant in the 19-year-old boy patient affected by cognitive regression and severe ID. Segregation analysis confirmed the heterozygosity state, not present in parents, and the mutation was confirmed by Sanger sequencing (data not shown). No other missense variants with CADD score > 15 or loss-of-function variants with a clear phenotypic association or compatible segregation pattern were identified (data not shown).

Specifically, the identified proband carries a single base pair deletion (NM_012431.3: c.621delG; p.R208Dfs*15) in the exon 6 of *SEMA3E* gene, which encodes for the functional SEMA domain of SEMA3E protein (Figure 1A). The deletion leads to a frameshift that generates a premature stop codon 15 amino acids downstream of the variant position, thus resulting in the production of a truncated protein lacking a large portion of SEMA domain and the entire PSI and Ig-like domains (Figure 1B).

Genomic coordinates on human GRCh37 genome assembly for the identified variant were chr7:83037733delC (Figure 1C). The variant, which was not found in 1000 genome project, ExAC, and gnomAD database (accessed on 3 April 2022), is predicted to be disease causing and likely pathogenic according to MutationTaster online an bioinformatics tool [28] and ACMG guidelines [29], respectively (Figure 1C).

Finally, the affected residue is partially conserved across species as per an average GERP score of 1.0075 (Figure 1D) and by multi-species alignment (Figure 1E). Of note, despite only primate and *Gallus gallus* orthologue proteins having a fully conserved R208 residue, in rodents this residue is replaced by a lysine, which belongs to the same chemical class of polar, positively charged amino acids, indicating conservation between amino acids with strongly similar properties. Instead, *Danio rerio* orthologue protein displays an asparagine at position 204, indicating conservation between amino acids of weakly similar properties.

### 2.3. The R208Dfs*15 Mutation Causes Altered Protein Localisation by Inducing Retention in the Endoplasmic Reticulum

To confirm the predicted pathogenicity of the identified *SEMA3E* variant (Figure 1), we performed cellular and biochemical assays. First, we tested if the R208Dfs*15 mutation affected protein localization. To achieve that, we overexpressed wild-type (WT) or mutant Alkaline Phosphatase (AP)-tagged human SEMA3E in COS-7 cells and applied an immunofluorescence protocol [30] to visualize protein localization using a previously tested anti-SEMA3E antibody [27]. Phalloidin staining was used to reveal actin cytoskeleton and overall cell morphology. As shown in Figure 2A–D, WT SEMA3E normally localized in the cytoplasm, as typical of secreted proteins. Instead, R208Dfs*15 SEMA3E mutant protein primarily localized in a perinuclear region, resembling the endoplasmic reticulum (ER) and thus suggesting a possible mechanism of protein retention and impaired secretion.

To verify the possible subcellular ER-retention of mutant SEMA3E, we performed co-localization experiments in COS-7 cells co-transfected with WT or R208Dfs*15 AP-SEMA3E and the mEmerald-ER-3 vector, used as a direct fluorescent ER marker. As shown in Figure 2E,F, mutated SEMA3E protein (red), detected with an anti-SEMA3E antibody, clearly and exclusively co-localized with the ER marker (green), while WT SEMA3E was also expressed in the cytosol. These findings confirmed that the mutation found in the patient affects protein localization and induces ER-retention, possibly impairing its physiological secretion.

### 2.4. The R208Dfs*15 Mutation Causes SEMA3E Truncation and Prevents Its Secretion and Binding to Immortalised Neurons

To assess the effect of the R208Dfs*15 mutation on SEMA3E production and secretion, we performed a Western blot (WB) analysis on cell lysates and conditioned media (CM) from COS-7 cells transfected for 24 and 48 h with WT or mutant SEMA3E. As shown in Figure 2G, the WB analysis on protein lysates revealed the presence of an expected band of ~165 KDa, corresponding to the SEMA3E monomer (95 KDa) and the AP-tag (70 KDa), for WT SEMA3E. Instead, and in agreement with in silico predictions, overexpression of mutant SEMA3E revealed a smaller band of ~95 KDa, corresponding to a short SEMA3E protein of 25 KDa plus the AP-tag (70 KDa), thus confirming the effect of the mutation to induce the formation of a truncated protein.

A similar WB analysis on CM from COS-7 cells transfected with WT or mutant SEMA3E revealed the normal presence of WT SEMA3E, but the complete absence of mutated SEMA3E. These results together with the localization studies confirmed that the patient’s frameshift mutation induces the formation of a truncated protein that is retained in the ER and is not secreted, thus strongly confirming the loss-of-function nature of the mutation.

To verify that R208Dfs*15 SEMA3E mutation induced ER retention and defective secretion, we assessed the ability of the CM from COS-7 cells transfected with AP-tagged WT or mutant SEMA3E to bind to the membrane of PLXND1-expressing GN11 cells, an immortalized cell line of immature neurons (Figure 2H–J), as previously described [27,31]. The conditioned media from COS-7 cells transfected with an empty AP-vector was used as negative control. As displayed in Figure 2L,M, WT SEMA3E-CM bound to the surface of GN11 cells, as revealed by the presence of a violet staining given by the deposition of the insoluble reaction product. Instead, R208Dfs*15 SEMA3E-CM failed to produce a colorimetric reaction, confirming absent secretion and consequent inability of the medium to bind to the surface of neurons.

### 2.5. The Enriched Conditioned Media of WT but Not Mutant SEMA3E, Binds to Embryonic Brain Tissue Expressing PLXND1

To further confirm that R208Dfs*15 mutation abolishes the physiological functions of SEMA3E during brain development, we mimicked the effects of this mutation with an ex vivo approach and performed similar binding assays on embryonic mouse brain tissues. First, by in situ hybridization and immunohistochemistry, we confirmed the expression pattern of *Sema3e* and PLXND1, respectively, on sections from WT mouse brain at embryonic day (E) 14.5 (Figure 3A–C).

We then extended expression studies on a human embryo at Carnegie Stage (CS) 19, corresponding to E12.5 in mouse (Figure 3D–F). As displayed in Figure 3B, *Sema3e* is expressed in several mouse brain regions such as the globus pallidus, the ventricular zone of the dorsal neocortex, and lateral ganglionic eminence, while PLXND1 is expressed in the striatum and piriform cortex, by cortical plate cells and subplate cells from dorsal to lateral cortical regions and by cells residing in the hippocampal formation and cortical hem (Figure 3C), as previously described [24,32]. Further, these experiments revealed that SEMA3E and PLXND1 are present in similar regions of the embryonic human brain (Figure 3D–F), strongly supporting a conserved biological role of SEMA3E/PLXND1 signaling in both mouse and human species.

Last, we exposed E14.5 fresh–frozen mouse sections to the CM of COS-7 transfected with WT or mutant SEMA3E. As shown in Figure 3G–I, while WT SEMA3E-CM bound to the brain of E14.5 mouse embryos, R208Dfs*15 SEMA3E-CM failed to react, as confirmed by the absence of violet precipitates. Altogether, these results strongly support a loss-of-function effect of the identified mutation, which impairs SEMA3E secretion and binding to its target cells during brain development.

## 3. Discussion

To the best of our knowledge, this is the first study identifying a de novo frameshift *SEMA3E* mutation in a patient affected by severe ID and cognitive regression.

Given the conserved expression pattern of SEMA3E and PLXND1 in mouse and human embryonic brain and our in vitro and ex vivo experiments confirming a loss-of-function effect of the human mutation, it is plausible to hypothesize that the clinical neurological features of our patient might be due, at least in part, to a defective SEMA3E signaling during embryonic brain development.

Although genetically engineered mice carrying the patient mutation would represent the closest model to study the effects of R208Dfs*15 variant ex vivo, our data suggest the mutation, by preventing protein secretion, is likely to abolish the physiological functions of SEMA3E during brain development, thus mimicking the brain defects observed in *Sema3e* knockout mouse models.

In this respect, the available phenotypic analyses of *Sema3e*-null mice revealed the presence of several neurodevelopmental defects that highly correlate with the clinical signs of our patient, which include worsening of his social, adaptive, and personal autonomy skills, cognitive regression according to scores on intelligence scales, ID, and tics.

Specifically, SEMA3E signaling was found to play a role during early development of descending axon tracts in the forebrain, including the corticofugal, striatonigral, and subiculo-mamillary tracts, which were mispatterned in *Sema3e*-null mice. As a consequence, adult *Sema3e*-null mice displayed several behavioral problems, including decreased anxiety and memory impairment [21], which are consistent with the patient phenotype.

In addition, SEMA3E signaling negatively regulates the migration of hem-derived Cajal-Retzius cells during early neocortical development. Accordingly, increased migratory capabilities of these cells were observed in adult *Sema3e*-null mice, resulting in an aberrant layering of the neocortex and consequent deficits in emotional behavior and working memory [24]. Considering that SEMA3E/PLXND1 is also a cell-intrinsic pathway regulating olfactory bulb newborn neuron migration [25], the R208Dfs*15 variant we identified might concurrently contribute to defective cortical layering, which could be linked with the cognitive regression of the patient.

Last, another study highlighted a role for SEMA3E signaling in the hippocampus, with *Sema3e*-null mice displaying abnormal entorhino–hippocampal connections, misrouted ectopic mossy fibers, and increased network excitability. Such changes in mossy fiber distribution have been shown to correlate with deficits in spatial and non-spatial memory [26], again in agreement with the patient’s clinical symptoms.

Interestingly, a previous work from our laboratory has also identified a role for SEMA3E in the survival of the hypothalamic GnRH neurons that control reproduction [27]. Further, the same work along with another paper [14] reported missense *SEMA3E* mutations in patients’ neurodevelopmental syndromes, such as Kallmann (KS) and CHARGE, both associated to GnRH deficiency and consequent reproductive defects. In this context, it will be interesting to expand *SEMA3E* mutational screenings in cohorts of patients with NDDs/ID or KS/CHARGE. This will help to understand the mechanisms underlying possible genotype–phenotype correlations and to study the biological relevance of missense versus non-sense mutations in the different neuronal subtypes affected by SEMA3E. This information will be vital to ameliorate diagnosis and therapeutic intervention of these disorders, which are still largely idiopathic.

In general, mutations in *SEMA3E* and in other members of the family, including neuropilin and plexin receptors, have so far mainly been linked to GD [8]. However, recent evidence, including this work, strongly supports that this class of molecules might also be implicated in broader NDDs [6,15,16,17,18].

Yet, the underlying genotypic–phenotypic correlations between SEMA mutations and different clinical manifestations are not known. Given the key role of SEMA signaling during several aspects of brain development [9], it will not be surprising if further genetic alterations in this class of molecules will emerge in patients affected by ID, whose genetic cause is still unknown.

Further studies are therefore needed to better understand the role of this family of genes in the susceptibility to NDDs. Such information will be vital to ameliorate diagnosis and therapeutic intervention of these disorders, which are still largely idiopathic.

## 4. Materials and Methods

### 4.1. Identification of Cases with Variants in SEMA3E Gene

511 cases of trio exome sequencing, collected in the Neurology Department of Hospital Universitario Quirónsalud Madrid since 2014, were analyzed to search for *SEMA3E* (Semaphorin 3E, NM_012431.3). Only loss-of-function mutations or variants with a CADD score >15 [33], in a heterozygosity state, according to the haploinsufficiency of this gene, were considered. All studies were performed on patients with neurodevelopmental disorders of probable genetic origin.

### 4.2. Neuroimaging

The brain MRIs performed at our center always entail different sequences, including DTI and 3D-MPRAGE T1, regardless of the reason for the study. DTI images were obtained with a 3T system (GE Medical System, Milwaukee, WI, USA) by using a SS-SE echoplanar Diffusion weighted image (DWI) sequence (TR: 12,000; FOV: 240 mm; sections thickness: 3 mm, 0 spacing; matrix 128 × 128; bandwidth: 250; 1 nex; diffusion encoding in 45 directions) with maximum b = 1000 s/mm^2^. Brain MRI analysis was conducted by a specialized radiologist who was unaware of the patient’s genetic diagnosis.

3D-tractography was performed in an off-line workstation by using commercially available processing software as provided by the manufacturer (Functool 3D Fiber Tracking, GE, Buc, France) based on fiber assignment by contiguous tracking (FACT) method, achieved by connecting voxel to voxel. The threshold values were 0.3 for FA and 45° for the trajectory angles, between the regions of interest (ROIs). DTI tracts were also co-registered to the 3D-T1 weighted data set. Before the contrast gadolinium-enhanced images, we included a prototype sequence to measure cortical perfusion called enhanced-ASL. This sequence was used with the following parameters: TE 2.8 ms; TR 4894 ms; post-labeling delay 2025 ms; bandwidth 62.5 kHz; field of view 22 cm; reconstructed image 128 × 128; and slice thickness 4 mm. Qualitative cortical perfusion maps were obtained in an off-line workstation with commercial software (Aw Server 3.2, GE, Buc, France).

The spectral technique used was the PRESS technique (Point Resolved Spectroscopy) and the parameters used were TE 37 ms, TR 3000 ms, 64 averages without H_2_O suppression, sweep width = 5000 Hz, and autoshimming. The voxel used had dimensions of 15 × 15 × 20 mm and is located in basal ganglia or corona radiata. Four to six saturation bands were used to increase homogeneity and avoid elements of confusion. Concentrations derived from the raw data obtained from the spectral curves in the MR were processed by the quantification program LCModel (S. Provencher), analyzing the spectra as a linear combination, based on a group of complete models of spectroscopies of metabolites in “in vitro” solution.

3D volumetric analysis for each patient was performed after brain MR imaging parcellation. Structural T1 weighted volumes were automatically segmented using FreeSurfer image analysis suite 7.1 (https://surfer.nmr.mgh.harvard.edu/ (accessed on 1 April 2022)), using default parameters. The result was a label map of isotropic voxel size (1 × 1 × 1 mm; 256 × 256 × 256 voxel) containing a plethora of brain regions, along with meaningful anatomical information: volume, area, and cortical thickness for each region. These results were compared with their control group, and matched for age and sex (normal pediatric and adult data set). As in previous studies with this technique, cold colors (blue) indicate less cortical thickness and warm colors (red) indicate greater cortical thickness vs. the control group.

### 4.3. Neuropsychological Assessment

A complete neuropsychological evaluation was performed on all patients presenting for neurodevelopmental disorders, depending on the patient’s age and medical history. In this case, we conducted a neuropsychological assessment to measure intellectual and language abilities, attention and executive functioning, and adaptive behavior. Specifically, the following tests and scales Spanish version tests were used, according to his age: Wechsler Preschool and Primary Scale of Intelligence, Third Edition, WPPSI-III [34]; Wechsler Intelligence Scale for Children, Fourth version, WISC-IV [35]; Wechsler Adult Intelligence Scale; Fourth version, WAIS [36]; Cumanin battery (Cuestionario de Madurez Neuropsicológica or Neuropsychological Maturity Questionnaire) [37]; Peabody Picture Vocabulary Test, Third Edition, PPVT-III [38]; Illinois Test of Psycholinguistic Abilities, ITPA [39]; Continuous Performance Test [40]; Test of Memory and Learning, Tomal-2 [41]; Rey complex figure test [42]; and Neuropsychological Evaluation of Executive Functions in Children, ENFEN (Evaluación Neuropsicológica de las Funciones Ejecutivas en Niños) [43]. Adaptative skills were evaluated with the Adaptive Behavior Assessment System II [44].

### 4.4. Genetic Analysis

Whole exome sequencing was performed using genomic DNA isolated from whole blood from proband and parents. Genomic DNA extraction was carried out from blood using the Magna Pure 24 equipment (Roche Diagnostics, Basel, Switzerland). Quantity of extracted gDNA was measured with a fluorimeter (Quibit 3.0). The absorbance ratios at 260/280 and 260/230 were also studied to determine the quality of the DNA obtained, using NanoDrop ND-2000 equipment. In addition, integrity of the genomic DNA was analyzed by electrophoresis in 0.8% agarose gels. Libraries were prepared using the KAPA Hyper plus Kit (Roche Diagnostics) following the manufacturer’s specifications and capture enrichment protocol with specific probes (KAPA HyperExome; Roche Diagnostics). Then, we performed subsequent massive parallel sequencing in a NextSeq550 equipment (Illumina, San Diego, CA, USA). Signal processing, base calling, alignment, and variant calling were performed with Genologica variant analysis software (GenoSystem, Swindon, UK). This software developed by Genologica contains an optimized algorithm that includes (among other steps) the following: (a) initial quality control of the sequences, (b) filtering the sequences by eliminating indeterminacies, adapters and low-quality areas, (c) second quality control of the sequences, (d) mapping on the Hg19 reference genome, (e) obtaining variants and CNVs, (f) mapping coverage study, and (g) annotating variants.

Finally, the prioritization variant was based on stringent assessments at both the gene and variant levels, and taking into consideration the patient’s phenotype and the associated inheritance pattern. Candidate variants were visualized using IGV (Integrative Genomics Viewer). Candidate variants were evaluated based on stringent assessments at both the gene and variant levels, taking into consideration both the patient’s phenotype and the inheritance pattern. Variants were classified following the guidelines of the American College of Medical Genetics and Genomics (ACMG) [29]. A board of molecular clinical geneticists evaluated each variant classified as pathogenic, likely pathogenic, or a variant of uncertain significance, and decided which, if any, had to be reported. In every case, causal variants were discussed with the referring physician and/or clinical geneticist.

### 4.5. Generation of Mutated SEMA3E Expression Vector

Site-directed in vitro mutagenesis was used to introduce the c.621delG mutation into an expression vector containing AP-conjugated human *SEMA3E* [21] using the QuickChange Lighting Kit (Agilent Technologies (Santa Clara, CA, USA) and the following oligonucleotides: 5′-ATGGGCCAGTCGCCCATGCTGCGG-3′, 5′-CCGCAGCATGGGCGACTGGCCCAT-3′.

### 4.6. Cell Lines

GN11 cells (gift of Dr. S. Radovick, University of Chicago, Chicago, IL, USA) and COS-7 cells (American Type Culture Collection, Manassas, VA, USA) were grown as a monolayer at 37 °C in a humidified 5% CO_2_ incubator in DMEM (Euroclone, Milano, Italy) supplemented with 10% fetal bovine serum (FBS, Life Technologies, Carlsbad, CA, USA), 1% L-Glutamine and 0.1% penicillin/streptomycin solution, referred as complete medium. Sub-confluent cells were harvested by trypsinization and cultured in 55 cm^2^ dishes at a density of 100,000 cells/dish for routine passaging.

### 4.7. Transfection Experiments

For transfection, COS-7 cells (at 80% confluence) were plated in standard 24-well plates at densities of 15,000 and 9000 cells/well to be fixed and immunostained 24 h and 48 h upon transfection, respectively. COS-7 cells were also plated at densities of 200,000 and 150,000 COS-7 cells/well in 6-well cell culture plates for the preparation of protein lysates and conditioned media 24 h and 48 h after transfection, respectively. COS-7 cells were grown in culture plates in complete culture medium for 24 h and transiently transfected upon incubation for 6 h with the selected expression vectors (1 µg/mL) in reduced-serum OPTIMEM medium (Gibco, Waltham, MA, USA) and in the presence of Lipofectamine 3000 (Invitrogen, Carlsbad, CA, USA), according to the manufacturer’s instructions. Six hours after, the transfection medium was replaced with complete medium for 24-well plates and high glucose DMEM without phenol red for 6-well plates. To visualize endoplasmic reticulum (ER), the mEmerald-ER-3 vector (gift from Michael Davidson, Addgene plasmid no. 54082) was co-transfected with the AP-SEMA3E vectors (ratio 1:4).

### 4.8. Immunoblotting

Cell lysates and conditioned media from COS-7 cells transiently transfected for 24–28 h with WT or c.621delG *SEMA3E* plasmids were obtained as previously described [27]. Briefly, conditioned media were collected, centrifuged at 13,000 rpm for 2 min and supernatant was used for analysis. Cells were lysed in 150 mM NaCl, 50 mM Tris- HCl (pH 7.4), and 1% Triton X-100, supplemented with protease and phosphatase inhibitors (Roche). Lysates were centrifuged at 13,200 rpm for 10 min at 4 °C and protein concentration determined with the Bradford assay (Bio-Rad, Hercules, CA, USA). Protein lysates (15 μg) or conditioned media (20 μL) were used for SDS-PAGE (8% polyacrylamide gels, under reducing conditions). Prestained Sharpmass VII (Euroclone) was used as protein molecular weight marker. Proteins were transferred to nitrocellulose membranes (Bio-Rad) and, after blocking with 5% non-fat milk in PBS containing 0.1% Tween-20 (PBS-T 0.1%) for 1 h at room temperature (RT), membranes were incubated overnight at 4 °C with primary antibodies: goat anti-SEMA3E (1:500; R&D Systems, Minneapolis, MN, USA); rabbit anti-glyceraldehyde-3-phosphate dehydrogenase (GAPDH, 1:1000, Cell Signaling, Danvers, MA, USA); followed by 1 h RT incubation with HRP-conjugated anti-goat and anti-rabbit antibodies (1:10,000; Santa Cruz Biotechnology Inc., Dallas, TX, USA), respectively. Detection was performed with enhanced chemiluminescence detection kit reagents (WASRAR ηC ultra-2.0; Cyanagen, Bologna, Italy).

### 4.9. RNA Extraction and RT-PCR

Total RNA was collected from GN11 cells and retrotranscribed to cDNA as previously described [45]. RT-PCR was performed using 50 ng of cDNA, the Quick-Load^®^ Taq 2X Master Mix (New England Biolabs, Ipswich, MA, USA) and specific primers for *Plxnd1* (FW 5′-TCCTAGACAGCCCTAACCCC-3′ and REV 5′-AGGCTCAATCGCTCGGATTT-3′) [27]. Amplification products were separated by 1% agarose gel electrophoresis and detected by ethidium bromide fluorescence on a UV transilluminator (Bio-Rad Laboratories).

### 4.10. Immunocytochemistry

Immunocytochemistry on COS-7 cells or GN11 cells was performed as previously described [30]. Primary antibodies were as follows: goat anti-human/mouse SEMA3E and goat anti-human/mouse PLXND1 (1:150; R&D Systems, Minneapolis, MN, USA). The following day, cells were incubated with Cy-3 or 488-conjugated donkey anti-goat (1:200; Jackson Immunoresearch, West Grove, PA, USA). Nuclei were counterstained with DAPI (1:10,000; Sigma, Burlington, MA, USA). To detect actin, cells were treated with phalloidin-TRITC (1:400, in PBS; Sigma) for 30 min at 37 °C before mounting with Mowiol (Calbiochem, San Diego, CA, USA).

### 4.11. Histology

Samples were dissected in PBS and fixed in 4% paraformaldehyde overnight at 4 °C and either cryopreserved in 30% sucrose for OCT embedding or, after dehydration, infiltrated and embedded in paraffin wax. Sagittal or coronal sections were cut at 20 μm using a cryostat or at 10 μm using a microtome.

### 4.12. In Situ Hybridization

Paraformaldehyde (PFA)-fixed cryosections were incubated with digoxigenin (DIG)-labeled anti-sense riboprobe for mouse *Sema3e* (Gift from Valerie Castellani). Hybridization step was performed in hybridization buffer (50% formamide, 0.3M sodium chloride, 20 mM Tris HCl, 5 mM EDTA, 10% Dextran sulphate, 1× Denhardt’s) overnight at 65 °C. Sections were washed with a series of gradually decreasing saline sodium citrate (SSC) buffers and then incubated overnight with AP-conjugated anti-DIG antibody (1:1500; Roche). mRNA expression was revealed by colorimetric staining using 4-Nitro blue tetrazolium chloride solution and 5-Bromo-4-chloro-3-indolyl phosphate disodium salt (NBT/BCIP, Roche).

### 4.13. Immunohistochemistry

PFA-fixed mouse cryosections (20 μm thickness) or paraffin-embedded human sections (10 μm) were incubated with hydrogen peroxide to quench endogenous peroxidase activity and then for 1 h at RT with PBS containing 10% normal horse serum and 0.1% TritonX-100. The following primary antibodies were used for immunostaining: goat anti-human/mouse SEMA3E or goat anti-human/mouse PLXND1 (1:50 and 1:200 respectively, R&D Systems). Sections were the incubated with biotinylated anti-goat antibody (1:400; Vector Laboratories, Burlingame, CA, USA) and then developed with the ABC kit (Vector Laboratories) and 3,3-diaminobenzidine (Sigma).

### 4.14. AP-Binding Assay

Human AP-SEMA3E proteins were prepared as previously described [31]. GN11 cells (60,000 cells/well) or fresh–frozen mouse E14.5 sections were fixed for 5 min in methanol at −20 °C, washed 5 times with PBS/MgCl_2_, incubated in PBS/MgCl_2_ containing 10% FBS for 1 h at RT and then reacted with the conditioned media from COS-7 cells transfected with WT AP-SEMA3E, mutated AP-SEMA3E or empty AP-vector) for 2 h at RT. Cells were then washed five times with PBS/MgCl_2_ and fixed with 4% PFA for 2 min at RT. Endogenous AP was heat inactivated by incubation at 65 °C for 3 h. Cell surface-bound, heat-stable recombinant AP activity was detected as an insoluble reaction product after incubation with nitro blue tetrazolium chloride and 5-bromo-4-chloro-3-indolyl phosphate (Roche).

### 4.15. Image Processing

Brightfield images were acquired using a Axiovert microscope (Zeiss, Jena, Germany). Immunofluorescence preparations were examined with an epifluorescent fluorescent microscope (Zeiss), and images were acquired with a Zeiss LSM900 laser scanning confocal microscope and a 40× objective (Zeiss Plan-Apochromat 40×, NA 1.3, Oil-immersion). ZEN 3.0 software (Zeiss) was used to process z-stacks at 0.25 μm intervals and generate maximal intensity projection images.

## Figures and Tables

**Figure 1 ijms-23-05632-f001:**
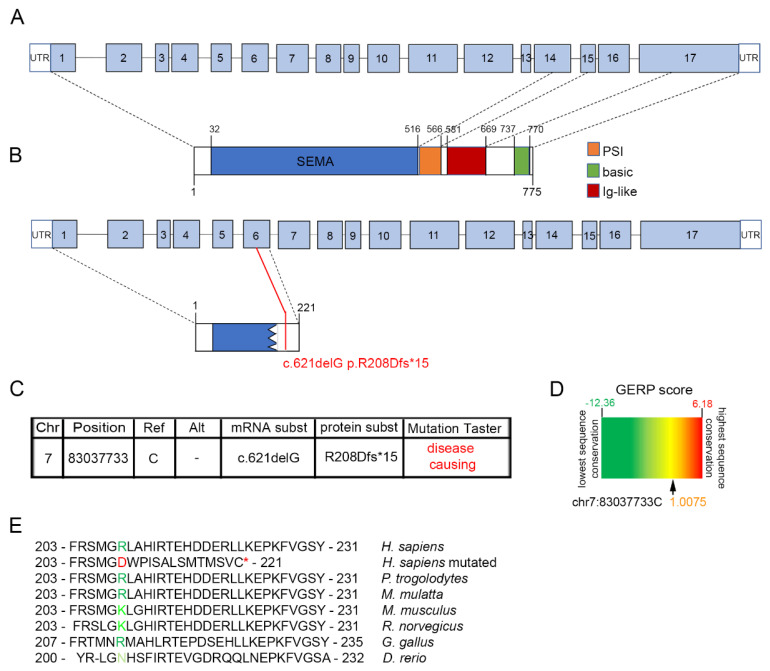
Identification of a de novo *Semaphorin 3E* (*SEMA3E)* frameshift variant in a patient with intellectual disability (ID). (**A**,**B**) Diagram of wild-type (WT) (**A**) and mutant (**B**) *SEMA3E* transcript and protein. (**C**) Chromosome position on human GRCh37 genome assembly, mRNA, and protein changes of the identified *SEMA3E* frameshift variant. The variant is predicted to cause disease with a probabilistic score above 0.99 [28]. (**D**) Genomic evolutionary rate profiling of sequence constraint for the mutated *SEMA3E* residue using GERP++ analysis provided a RS score of 1.0075, which indicates an intermediate level of conservation across all mammalian species. (**E**) Alignment of partial protein sequences of indicated vertebrate SEMA3E orthologues shows that the R208 residue is evolutionarily conserved in most species (dark green). *Homo sapiens* mutated residue is labeled in red; the asterisk indicates a premature stop codon. Abbreviations: Chr, chromosome; Ref, reference; subst, substitution.

**Figure 2 ijms-23-05632-f002:**
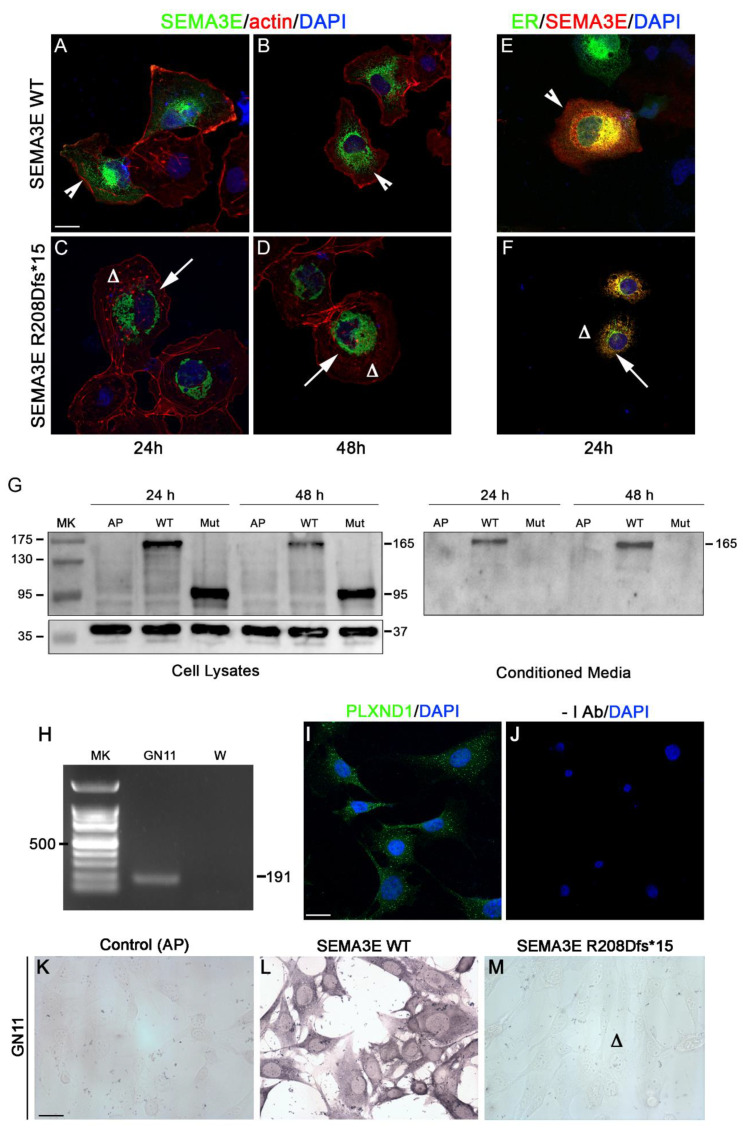
The R208Dfs*15 frameshift mutation causes defective SEMA3E localization, secretion, and binding. (**A**,**B**) Immunofluorescence for SEMA3E (green) on COS-7 cells at the indicated time-points revealed that WT SEMA3E localizes in the cytoplasm (white arrowheads), (**C**,**D**) whereas mutant SEMA3E localized in a restricted perinuclear region (white arrows). Cytoskeletal actin was visualized with TRITC-conjugated phalloidin (red). (**E**,**F**) Immunofluorescence for SEMA3E (red) and endoplasmic reticulum (ER) (green) on COS-7 cells revealed a partial co-localization of WT SEMA3E with the ER (E; white arrowhead) and an exclusive co-localization of the mutant (F; white arrow), which was not detected in the rest of the cytoplasm (F; ∆). Nuclei were counterstained with DAPI (blue). Scale bars: 25 μm. (**G**) Western blot for SEMA3E on cell lysates from transfected COS-7 revealed the presence of bands of ~165 kDa for the WT protein and of ~95 kDa for mutated SEMA3E (mut), indicative of a truncated protein. GAPDH (37 kDa) was used as loading control. Immunoblotting of SEMA3E on conditioned media (CM) revealed that the R208Dfs*15 mutation prevents SEMA3E protein secretion as confirmed by absent bands. (**H**–**J**) PlexinD1 (PLXND1) expression in GN11 cells is confirmed by RT-PCR (**H**) and immunofluorescence (**J**). No primary antibody was used as negative control. W, water. Scale bars: 100 μm. (**K**–**M**) Alkaline Phosphatase (AP)-binding assay on GN11 cells with CM from transfected COS-7 cells revealed the binding of WT (**K**), but not of mutant (**L**) SEMA3E. CM from COS-7 transfected with an empty vector was used as negative control (**M**). Scale bars: 100 μm.

**Figure 3 ijms-23-05632-f003:**
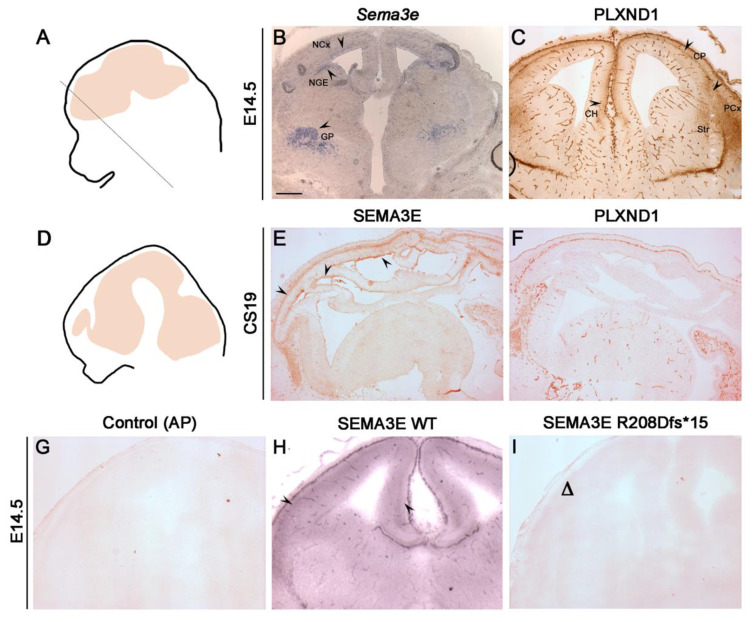
SEMA3E and PLXND1 are expressed in the human/mouse embryonic brain and human SEMA3E binds to embryonic mouse brain ex vivo. (**A**) Schematic representation of a coronal section plane of a mouse embryonic day 14.5 (E14.5) head (dotted line). (**B**) In situ hybridization for *Sema3e* on E14.5 mouse sections revealed expression in the globus pallidus (GP), ventricular/subventricular zone of the dorsal neocortex (NCx), and ventricular zone of the lateral ganglionic eminence (LGE) (arrowheads). Scale bar: 500 μm. (**C**) Immunohistochemistry on E14.5 mouse sections for PLXND1 showed its expression in the striatum (Str), piriform cortex (PCx), cortical plate (CP) cells, and subplate cells from dorsal to lateral cortical regions (arrowheads). Some expression was also observed in the hippocampal and dentate neuroepithelium and cortical hem (CH). Scale bar: 500 μm. (**D**) Schematic representation of the sagittal section plane of an embryonic human head at Carnegie stage (CS) 19. (**E**,**F**) Immunohistochemistry on sections from CS19 human head for SEMA3E (**E**) and PLXND1 (**F**) revealed that both are expressed in the CP. SEMA3E is also expressed in ventricular regions (arrowheads). Scale bar: 500 μm. (**G**–**I**) AP-binding assay on E14.5 coronal mouse sections showed binding at the level of CP and CH (arrowheads) only upon exposure to CM containing WT protein (**G**), whereas mutant SEMA3E did not show any binding (**H**), as was shown with the negative control (**I**). Scale bar: 500 μm.

**Table 1 ijms-23-05632-t001:** Summary of the patient’s clinical features at the age of 5 years. Abbreviations: WPPSI-III, Wechsler Preschool and Primary Scale of Intelligence III; MRI, Magnetic Resonance Imaging.

Clinical Features And Studies	
Age	5 years
Weight	19 kg (35th centile)
Height	110 cm (30th centile)
Occipital-frontal diameter	51 cm (50th centile)
Communication deficits	Verbal and nonverbal
Neuropsychological evaluation	Intellectual Quotient: 83 (WPPSI-III) Developmental Quotient: 70 (CUMANIN)
Neurometabolic tests	Normal: blood count, blood biochemistry, hepatorenal function, thyroid hormones, ammonium, lactic, pyruvic, amino acids and organic acids in blood/urine.
Brain 3T MRI	Normal
Genetic studies	Normal: Karyotype and 400k aCGH

**Table 2 ijms-23-05632-t002:** Worsening of the patient’s IQ scores going from childhood to adolescence. Scores have been measured according to Weschler Scales (WPPSI-III, WISC-IV, and WAIS).

Age (Years)	IQ Score
5	83
7	63
10	50
19	38

## Data Availability

Data describing the identified variant have been submitted to ClinVar Database (https://www.ncbi.nlm.nih.gov/clinvar/): ClinVar Submission number SUB11041179.

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
