# Peer review of "A Novel Loss-of-Function SEMA3E Mutation in a Patient with Severe Intellectual Disability and Cognitive Regression"

_ijms, 2022, doi:10.3390/ijms23105632_

Round 1

Reviewer 1 Report

First of all, I would like to congratulate the authors for their great work. I think it is an exemplary work. I have only minor considerations to make to broaden the number of possible readers and make it specifically more interesting for clinicians: 
1) In the introduction point out that not all causes of intellectual disability are genetic. Some mention of other prenatal infectious causes, peripartum complications... I think it would be good. Also pointing out that the most frequent cause of intellectual disability is Down syndrome would also be more appropriate. 
2) In the introduction, some mention of the classification of mild/Moderate/Severe/Profound intellectual disability would be good. Not everyone has to know what is the borderline between the different levels of intellectual disability (classification based on DSM-V or ICD-10 criteria?). Add a comment. 
3) Methodology section: I think it could make more sense before the results but it is a personal opinion. What I would advise is to add information on the methodology used for neuropsychological and medical evaluation (scales used adding the relevant bibliographic references justifying their use), on brain MRI (3T or 1.5 T and protocol used), physical examination findings (changes in visual acuity? hearing? body dysmorphias? throughout puberty/early adulthood). 
4) It would also be advisable to explain the concept of regression more clearly. It is an entity that is being studied in the context of different intellectual disabilities such as Down syndrome. In this context there is usually an emotional and/or medical trigger that justifies it most of the times. Do treatments postulated for other cases of intellectual disability that present possible regression make sense in this context? Do symptomatic treatments have little benefit in general but would they make sense here? What could they provide?
5) Adding a table or figure summarizing the clinical picture could also increase the quality and interest of the manuscript. 
Beyond these minor comments, nothing more than congratulations to the authors for the work done.

Author Response

Response to Reviewer 1 Comments
First of all, I would like to congratulate the authors for their great work. I think it is an exemplary work.
I have only minor considerations to make to broaden the number of possible readers and make it
specifically more interesting for clinicians:

We thank the reviewer for his/her positive view on our work and the constructive comments that have
been totally considered in the revised manuscript. See detailed response to the specific points:

Point 1: In the introduction point out that not all causes of intellectual disability are genetic. Some
mention of other prenatal infectious causes, peripartum complications... I think it would be good. Also
pointing out that the most frequent cause of intellectual disability is Down syndrome would also be
more appropriate.

Response 1: thanks for pointing out this important aspect, that was taken into consideration and now
inserted in the introduction (Pag 1, lines 41-51).

Point 2: In the introduction, some mention of the classification of mild/Moderate/Severe/Profound
intellectual disability would be good. Not everyone has to know what is the borderline between the
different levels of intellectual disability (classification based on DSM-V or ICD-10 criteria?).

Response 2: thanks for pointing out this important aspect. We have now inserted in the Intorduction
the classification of ID based on DSM-5 (Pag 1, lines 39-41).

Point 3: Methodology section: I think it could make more sense before the results but it is a personal
opinion. What I would advise is to add information on the methodology used for neuropsychological
and medical evaluation (scales used adding the relevant bibliographic references justifying their use),
on brain MRI (3T or 1.5 T and protocol used), physical examination findings (changes in visual acuity?
hearing? body dysmorphias? throughout puberty/early adulthood).

Response 3: regarding the order of the sections, this was dictated by the Journal’s guidelines,
therefore we cannot move up the Methodology section. Regarding the need of adding further
information on the methodology used for neuropsychological and medical evaluation, we have now
inserted more details in this section according to the Reviewer suggestion (Pag.10-12, lines 370-422).

Point 4: It would also be advisable to explain the concept of regression more clearly. It is an entity
that is being studied in the context of different intellectual disabilities such as Down syndrome. In this
context there is usually an emotional and/or medical trigger that justifies it most of the times. Do
treatments postulated for other cases of intellectual disability that present possible regression make
sense in this context? Do symptomatic treatments have little benefit in general but would they make
sense here? What could they provide?

Response 4: we thank the Reviewer for raising these important aspects, that have now been
explained better in the Results section (Pag.4, table 2B).

Point 5: Adding a table or figure summarizing the clinical picture could also increase the quality and
interest of the manuscript.

Response 5: as suggested by the Reviewer, we have now included in the Results section (page.4,
table 2A) a table to summarise the clinical features presented by the patient to increase quality and
interest of the manuscript.

Beyond these minor comments, nothing more than congratulations to the authors for the work done.

Response: thank you once again for his/her positive view.

Reviewer 2 Report

The authors demonstrate the pathogenic impact of a de novo deletion leads to a frameshift that generates a premature stop codon 15 amino acids downstream of the variant position. They provide evidence that clinical neurological features of the patient might be due to a defective SEMA3E signaling in the brain. Nowadays ~10% of detected variants by next generation sequencing are classified as variants of uncertain significance (VUS), a term indicating that there are insufficient data to distinguish between disease-causing mutations and benign variants. These undiagnosed patients represent an unsolved need, located on the frontier of the translational process between research and the clinical practice. Determining the phenotypic consequences of these variants is a major goal for genomic medicine. In addition, the genetic diagnosis of rare diseases through the exome or genome sequencing is made difficult by the identification of variants in genes whose involvement in the phenotype is not fully determined. When searching for the SEMA3E gene in OMIM, it is provisionally related to Charge syndrome. The professional HGMD database has collected 14 missense variants of this gene, 2 responsible for Charge syndrome, 1 associated with Kallmann syndrome and another with multiple congenital anomalies. The other 10 variants are presented as possibly responsible for other pathologies such as epileptic encephalopathy, microphthalmia and coloboma, severe obesity, among others phenotypes. In these cases, the genetic diagnosis is complicated, since there is no well-established genotype-phenotype correlation.

The authors demonstrated the pathogenic impact of the variant and provide evidences that a defective SEMA3E signaling in the brain could be responsible for the clinical manifestations in the patient.

Author Response

Reply: We thank this reviewer for appreciating the importance of our work.